# Relevance of the Scaphoid Shift Test for the Investigation of Scapholunate Ligament Injuries

**DOI:** 10.3390/jcm11216322

**Published:** 2022-10-26

**Authors:** Daniel Schmauss, Sebastian Pöhlmann, Andrea Weinzierl, Verena Schmauss, Philipp Moog, Günter Germann, Berthold Bickert, Kai Megerle

**Affiliations:** 1Division of Plastic Surgery and Hand Surgery, Klinikum Rechts der Isar, Technische Universität München, Ismaninger Str. 22, 81675 München, Germany; 2Department for Plastic, Reconstructive and Esthetic Surgery, Ospedale Regionale di Lugano, Via Tesserete 46, 6900 Lugano, Switzerland; 3Faculty of Biomedical Sciences, Università Della Svizzera Italiana, 6900 Lugano, Switzerland; 4Department for Hand, Plastic and Reconstructive Surgery, Burn Center—BG Trauma Center Ludwigshafen, Plastic and Hand Surgery of the University of Heidelberg, Ludwig-Guttmann-Straße 13, 67071 Ludwigshafen am Rhein, Germany; 5Department of Plastic Surgery and Hand Surgery, University Hospital Zurich, Rämistrasse 100, 8006 Zurich, Switzerland; 6Ethianum, Clinic for Plastic, Reconstructive and Aesthetic Surgery, Voßstraße 6, 69115 Heidelberg, Germany; 7Department for Plastic Surgery and Hand Surgery, Schön Klinik München Harlaching, Balanstraße 71a, 81541 München, Germany

**Keywords:** scaphoid shift test, Watson test, scapholunate ligament, radial-sided pain, arthroscopy

## Abstract

**Background:** Although it is part of the common clinical examination of scapholunate ligament pathologies, there are only little data on the diagnostic value of the scaphoid shift test. The aim of this study was to evaluate the scaphoid shift test in a large cohort of patients. **Materials and Methods:** We retrospectively analysed 447 patients who underwent the scaphoid shift test and wrist arthroscopy because of various suspected injuries of the wrist, correlating the results of clinical examination with data obtained during the wrist arthroscopy. Sensitivity, specificity, and positive and negative predictive values were calculated and evaluated. **Results:** The sensitivity of the scaphoid shift test was low (0.50) when examining the whole cohort. In a subgroup of patients specifically referred for suspected scapholunate ligament injury, the sensitivity was higher (0.61), but the specificity was low (0.62). In detecting more serious lesions (Geissler 3 + 4), the scaphoid shift test demonstrated higher sensitivity (0.66). **Conclusions:** An isolated scaphoid shift test may only be of limited value in the diagnosis of scapholunate ligament lesions and should, therefore, be viewed as a useful tool for a preliminary assessment, but a negative test should not prevent the surgeon from indicating a more extensive diagnostic workup.

## 1. Introduction

The scapholunate ligament (SLL), an important primary stabilizer of the wrist, is the most commonly injured intercarpal ligament [1,2,3,4,5].

Lesions of the SLL and other secondary stabilizers (e.g., the radioscaphocapitate ligament, long and short radiolunate ligament) lead to scapholunate dissociation (SLD), which can be dynamic or static [4]. A dynamic SLD is present when the SL gap is asymmetrically widened on stress X-rays, while it is normal on neutral, static X-rays. In the case of a static SLD, the asymmetrically widened SL gap is already visible on neutral, static X-rays [6].

SLD is probably the most common cause for carpal instability. It is mostly of traumatic origin, with the most common traumatic mechanism being a fall on the dorsally extended hand [7].

Today, scapholunate instability is seen as a spectrum of injuries rather than an “all-or-nothing” phenomenon [8,9]. As a result of a rotary subluxation of the scaphoid, Watson described four types of instability, using static and dynamic X-rays [10,11]: predynamic instability, dynamic instability, static scapholunate dissociation, and scapholunate advanced collapse (SLAC). The X-rays of a predynamic instability do not show any malalignment, while the scapholunate gap in stress X-rays is widened in a dynamic instability, even though the scapholunate joint maintains its alignment when no stress is applied due to secondary stabilizers. While static instabilities demonstrate a dorsal intercalated segment instability (DISI) configuration in standard lateral views without loading, dynamic instabilities do not.

In the case of a static scapholunate dissociation, the malalignment due to the complete discontinuity of the SLL is already evident in static X-rays. If the wrist shows evidence of arthritis due to the displacement of the scaphoid and the resulting malalignment, Watson categorizes the injury as a SLAC lesion, with its own spectrum and grading based on the arthritic damage, the end stage being arthritic damage throughout the radiocarpal and midcarpal joints.

The clinical diagnosis of SLL lesions is still a challenge for hand surgeons, especially isolated lesions can be hard to detect. This is of clinical significance as the early diagnosis of SLD is important for an adequate therapy and the prevention of secondary damage [12].

Apart from obtaining a detailed patient history, the clinical examination may provide important information in diagnosing wrist pain in general and SLL lesions in particular. Radial wrist pain, tenderness in the anatomical snuffbox, and a history of trauma may suggest SLL injuries. Further radiographic workup consists of neutral and stress views in order to detect and differentiate dynamic and static SLD. Arthroscopy of the wrist as the final step of the diagnostic workup still represents the gold standard in detecting SLL lesions, which are arthroscopically classified according to Geissler in grades 1–4 [13].

The scaphoid shift test or “Watson test” is a widely used clinical test that can be used for the clinical diagnosis of SLL instability since the manoeuvre provokes a subluxation of the scaphoid over the dorsal rim of the radius [12,14,15,16].

However, the data in literature concerning the diagnostic value of the scaphoid shift test for the detection of SLL lesions are insufficient due to small patient cohorts. Therefore, the purpose of this study was to examine the relevance of the scaphoid shift test and, in particular, the specificity for the detection of SLL lesions in a large cohort of patients.

## 2. Materials and Methods

**Patients:** Between January 2001 and January 2006, we performed 1134 arthroscopies in 1103 patients referred to our hospital (Figure 1).

We reviewed the records of all patients, and we excluded 220 patients with 233 arthroscopies from further analysis due to an age under 18 years (40 patients), congenital variations of the wrist (44 patients), or established degenerative or post-traumatic arthritis (136 patients).

In the remaining patient cohort of 883 patients, the scaphoid shift test was preoperatively performed and the result was documented in 447 patients who underwent 452 arthroscopies. The mean age of this cohort was 38.7 years (range from 18 to 68 years) with 171 female and 276 male patients.

All patients were referred to our institution because of wrist pain. A total of 377 patients had a history of trauma to the wrist (fall to the extended hand). The mean delay from the trauma or the onset of symptoms to the arthroscopy was 13.6 weeks (range 1–120 weeks).

We extracted the results of the scaphoid shift test from preoperative records. Since the study cohort also included patients who were referred to our institution specifically for suspected SLL injury, we formed a more homogeneous subgroup of 213 arthroscopies in 211 patients and analysed it separately. The mean age of those patients was 38.0 years (range 18 to 61 years) with 65 female and 146 male patients. The mean delay in this subgroup from the trauma or the onset of symptoms to the arthroscopy was 13.5 weeks (range 1–72 weeks).

In those two cohorts, we further analysed the specificity of the scaphoid shift test depending on the severity of the SLL injury found during the arthroscopy according to Geissler (grades 1 + 2 vs. 3 + 4) [13].

Additionally, we conducted an analysis of the whole cohort in which only static SLL lesions were defined as “SLL injury”, while no lesion or dynamic lesions were defined as “no SLL injury”.

This study was conducted in accordance with the ethical guidelines of our institution.

**Scaphoid shift test:** The scaphoid shift test was conducted as follows, according to Watson [14]:

The examiner and patient sit face-to-face across a table with diagonally opposed hands raised (left to left or right to right) and elbows resting on the table between them. The examiner places his thumb on the palmar prominence of the scaphoid and wraps his fingers around the distal radius. The examiner’s other hand stabilizes the metacarpal level, thereby controlling the wrist position (Figure 2A). Under constant thumb pressure, the ulnar-deviated, slightly extended wrist is moved towards the radial side and into slight flexion (Figure 2B). If the scaphoid (painfully) subluxates, the examiner might notice a palpable or audible “clunk” when releasing thumb pressure. The test is then valued as ‘positive’, especially if there is a difference to the contralateral hand.

**Arthroscopy:** We collected all pathologies found during the arthroscopy in all patients. All arthroscopies were performed under standardized conditions with the wrist extended by a 5 kg distraction. For the visualization of the SLL, an arthroscope with a 30° view angle (Storz, Tuttlingen, Germany) was inserted through the 3/4 portal in all cases, and probes were introduced through the 4/5 portal. Additional evaluation of the SLL was performed via MCR and MCU portals in 329 cases. The SLL could be visualized in all cases and was further tested for lesions by inserting a probe. Various surgeons (>20) at all levels of training performed the procedures. However, a board-certified hand surgeon was present during all operations and interpreted the findings in each case. We collected all arthroscopic findings during the operation in a database directly connected to the arthroscopy unit. The arthroscopic diagnoses and Geissler classifications of all SLL lesions were later obtained from this database and correlated to the clinical test results documented in the patient records.

**Statistics:** Sensitivity, specificity, positive predictive values (PPV), and negative predictive values (NPV) were calculated using a fourfold table (Figure 3).

## 3. Results

### 3.1. Analysis of the Whole Cohort

Figure 4 summarizes the results for the whole cohort. The analysis of all 447 patients showed a PPV and NPV of the scaphoid shift test of 0.54 and 0.75, respectively. The sensitivity of the scaphoid shift test was 0.50, and the specificity was 0.78.

When SLL injuries were subdivided into categories grade 1 + 2 vs. grade 3 + 4 according to the Geissler classification, the sensitivity of the scaphoid shift test to detect the more serious injuries (Geissler 3 + 4) was 0.66 and, therefore, higher compared to the detection of injuries classified as Geissler grade 1 + 2 (0.27) (Figure 5). In one patient, a SLL injury was described in the operation report without specifying the Geissler classification. The patient could, therefore, not be included in this subgroup analysis.

### 3.2. Patients with Suspected SLL Injury

Figure 6 summarizes the subgroup analysis of patients who were specifically referred to the clinic for a suspected SLL injury. The PPV (0.65) and the sensitivity (0.61) were higher, and the NPV (0.58) and the specificity (0.62) were lower when compared to the values of the whole collective.

In the subgroup of patients referred for suspected SLL injuries, we compared patients with SLL injuries classified as Geissler grade 1 + 2 vs. grade 3 + 4. Again, the sensitivity of the scaphoid shift test was higher in the detection of severe injuries (0.72 vs. 0.38) (see Figure 7). 

### 3.3. Static Lesions vs. No Lesion/Dynamic Lesion

In Figure 3, we defined dynamic as well as static SL lesions as ”SL ligament injured”. However, in Figure 8, we evaluated static lesions separately. Therefore, Figure 8 demonstrates the analysis of the whole collective with static SLL lesions defined as “SL ligament injured”, while no lesions or dynamic lesions during the arthroscopy were defined as “SL ligament healthy”. The sensitivity (0.59) and specificity (0.75) were comparable to the analysis of the whole collective in Figure 3, while the PPV (0.35) was lower and the NPV higher (0.89).

## 4. Discussion

The scaphoid shift test is a commonly used clinical test for the diagnosis of scapholunate instability [3,14,16]. The aim of the present study was to evaluate the diagnostic significance of this test in a large cohort of patients in a clinical setting. Our data demonstrate that the scaphoid shift test has a limited diagnostic value for lesions of the SLL with a low PPW of 0.54 and a sensitivity of 0.50. The diagnostic value seems to depend, to some extent, on the severity of the lesion and the degree of the injury, since a subcohort of patients with higher-grade injuries of the SLL showed a higher sensitivity.

To the best of our knowledge, only three studies with smaller populations have been conducted, comparing clinical examinations against the arthroscopy findings and investigating the diagnostic value of the scaphoid shift tests in the detection of SLL injuries in particular [17,18,19]. 

LoStyo et al. analysed the sensitivity, specificity, and predictive values of the scaphoid shift test in 50 painful wrists and compared the test result to the arthroscopy findings [17]. The authors report a sensitivity of 69%, a specificity of 66%, a PPV of 48%, and an NPV of 78% and conclude that the test might be valuable to indicate a more detailed diagnostic workup.

Prosser et al. conducted a cross-sectional study and evaluated the diagnostic accuracy of different provocative wrist tests, amongst others, the scaphoid shift test, in 105 patients admitted with wrist pain and suspected wrist injuries and compared the findings to the arthroscopy [18]. The authors found a positive likelihood ratio of 2.88 and a negative likelihood ratio of 0.28 for the scaphoid shift test and classified this provocative test as “mildly useful” for both positive and negative test results.

Ruston et al. evaluated the diagnostic value of clinical examination and magnetic resonance imaging for common wrist pathologies in 66 patients who underwent arthroscopy [19]. Regarding the scaphoid shift test, the authors found a sensitivity of 47.6%, a specificity of 66.7%, a PPV of 40%, and an NPV of 73.2%. Due to these results that might indicate a limited diagnostic value, the authors suggested that magnetic resonance imaging should be performed for further workup if the scaphoid shift test result is positive.

These three studies concede that the scaphoid shift test has only a limited diagnostic value and are, therefore, in line with the findings of the present study.

The reason for the limited ability of the test to detect a SLL lesion may be that healthy individuals with a naturally lax connective tissue can show a positive test, while lower grade lesions do not necessarily lead to a positive scaphoid shift test [20]. Watson himself described the test as less of a test with a binary outcome but more of a provocative manoeuvre, stating that high mobility should not be considered pathological [15]. Some surgeons, therefore, believe that the scaphoid shift test should not be called a test but rather a sign.

Easterling and Wolfe performed the scaphoid shift test in asymptomatic patients and reported a prevalence of a positive scaphoid shift test in up to 32% [21]. Positive tests were associated with a general pronounced ligament laxity of the patient [21,22,23]. Interestingly, Wolfe et al. found an asymmetry between the two wrists in 14% of patients. An asymmetry between the wrists should, therefore, not be considered categorically pathological but can be an important factor when evaluating the clinical findings of a positive scaphoid shift test.

Another factor that should be taken into consideration is the presence or absence of pain during the scaphoid shift test. Park demonstrated in his study that abnormal mobility of the scaphoid is associated with pain and concluded that the scaphoid shift test should only be considered positive when it is associated with pain [24].

None of the above-mentioned studies evaluated the diagnostic value of SLL lesions in correlation to the Geissler classification. In the present study, we could clearly demonstrate a higher diagnostic value of the scaphoid shift test for higher-grade lesions, with a sensitivity of 0.66 for Geissler 3 and 4 lesions—more than twice as high as the sensitivity for Geissler 1 and 2 lesions (0.27). We conclude from these findings that a potentially negative test result should be interpreted with caution in cases with less pronounced symptoms in order to not miss lower-grade SLL lesions, since the sensitivity of the scaphoid shift test for the detection of low-grade lesions might be low. Therefore, the threshold to indicate arthroscopy should remain low in symptomatic patients, even if the scaphoid shift test is negative. However, it has to be acknowledged that low-grade lesions without signs of radiographic instability may not require surgical treatment. We feel that “predynamic” instabilities are not instabilities by definition and, therefore, suggest that the term “predynamic instability” should be abandoned.

Our study has several limitations, starting with the retrospective design and the fact that only patients with positive radiologic findings or the clinical suspicion of some form of wrist pathology underwent wrist arthroscopy. Conversely, most patients who had a negative result in the scaphoid shift test did not undergo arthroscopy, resulting in a possible bias of the negative predictive value and specificity. For obvious ethic reasons, no arthroscopies were performed in asymptomatic patients, and a comprehensive study cohort is, therefore, probably impossible to obtain. Another limitation of our study is the fact that midcarpal arthroscopy was not routinely performed on all patients. We generally feel that visualization of the SLL is more reliable from the midcarpal joint space, but the retrospective nature of this study makes it difficult to assess why, in some cases, only radiocarpal arthroscopy was performed. It cannot be ruled out that some SLL injuries might have been missed, but considering the large total number of cases, we do not think that this would alter the outcome of this study substantially.

The fact that more than 20 different surgeons at varying levels of training conducted the clinical examinations and the arthroscopies should also be considered a possibly confounding variable. However, all surgeons were undergoing the same program of surgical training, and a board-certified hand surgeon was present in each case.

As the test result is binary, it cannot adequately reproduce the spectrum of findings Watson himself described [15].

## 5. Conclusions

Based on the findings of this study, we believe that the scaphoid shift test should be considered a useful tool during the clinical examination, but a negative test in a symptomatic patient should never prevent the surgeon from indicating a more extensive diagnostic workup if the suspicion of a wrist injury persists, since the test becomes less reliable with lower-grade injuries.

## Figures and Tables

**Figure 1 jcm-11-06322-f001:**
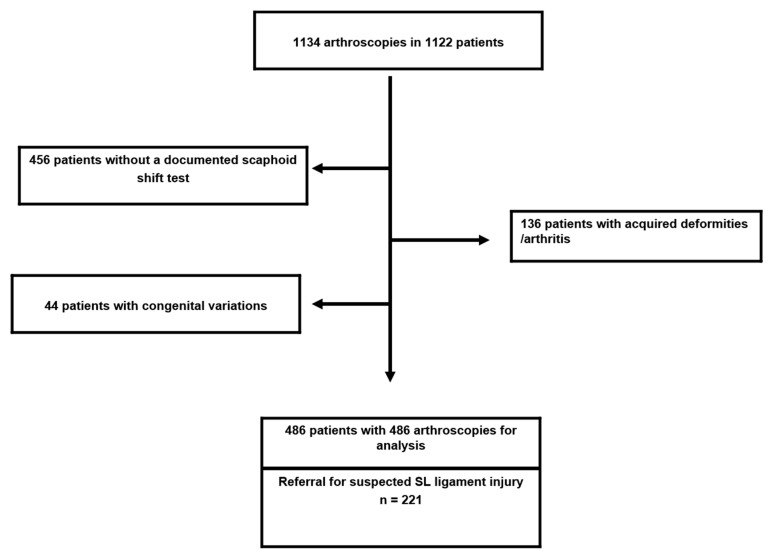
Illustration of the cohort of patients.

**Figure 2 jcm-11-06322-f002:**
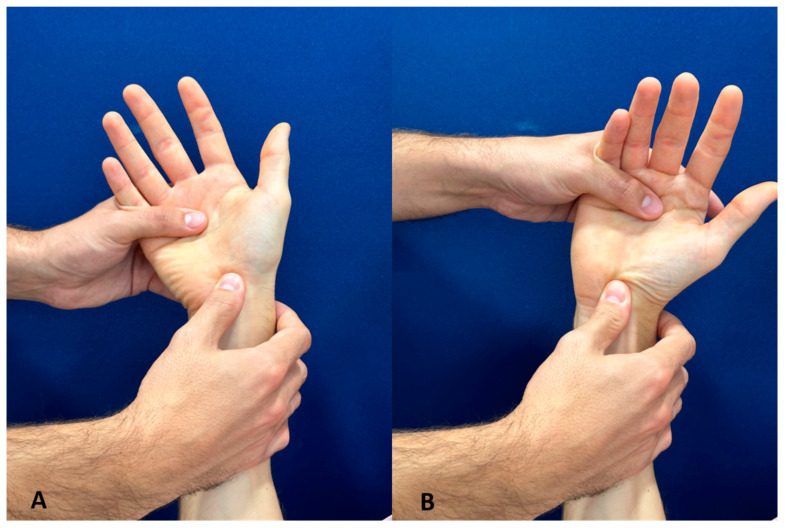
The scaphoid shift test: The examiner places his thumb on the palmar prominence of the scaphoid; the wrist is ulnar-deviated and slightly extended (**A**). The wrist is moved towards the radial side and into slight flexion (**B**).

**Figure 3 jcm-11-06322-f003:**
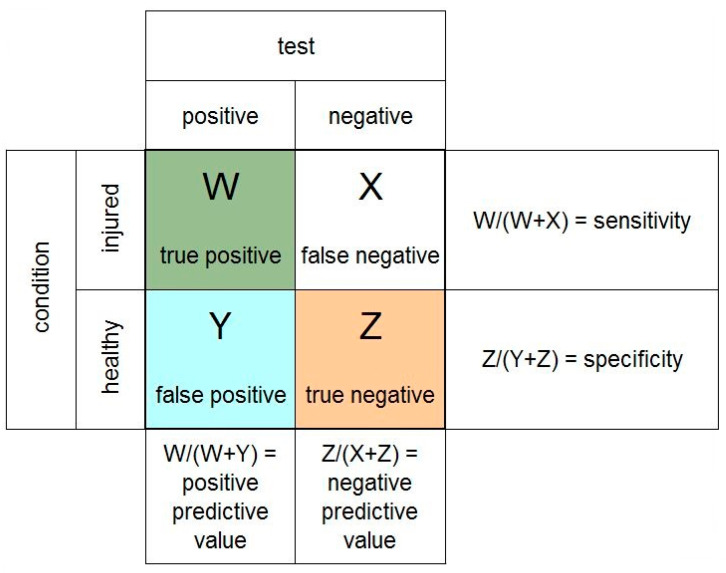
Model of exemplary fourfold table.

**Figure 4 jcm-11-06322-f004:**
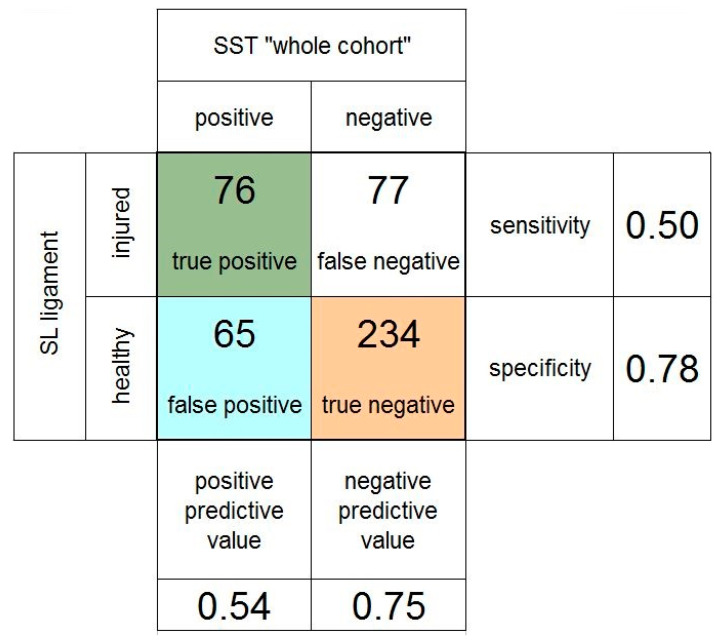
Fourfold table of the scaphoid shift test (SST) in the whole cohort of patients. SL: scapholunate.

**Figure 5 jcm-11-06322-f005:**
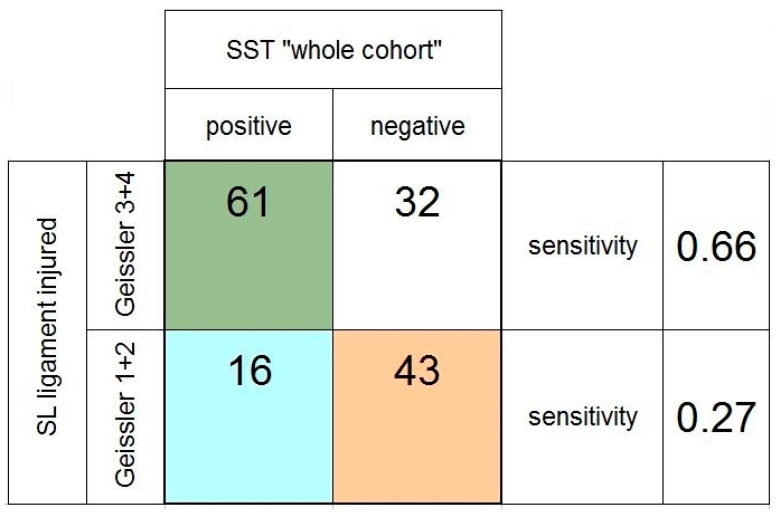
Fourfold table of the scaphoid shift test (SST) in the whole cohort of patients, depending on the severity of the scapholunate (SL) ligament injury (grade 1 + 2 vs. grade 3 + 4, according to the Geissler classification).

**Figure 6 jcm-11-06322-f006:**
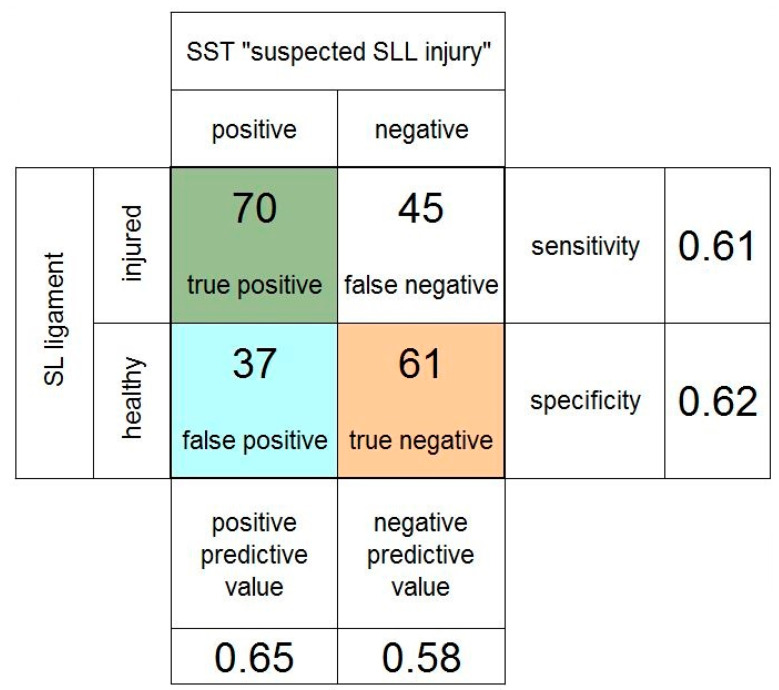
Fourfold table of the scaphoid shift test (SST) in the subgroup of patients referred specifically for suspected scapholunate (SL) ligament injury.

**Figure 7 jcm-11-06322-f007:**
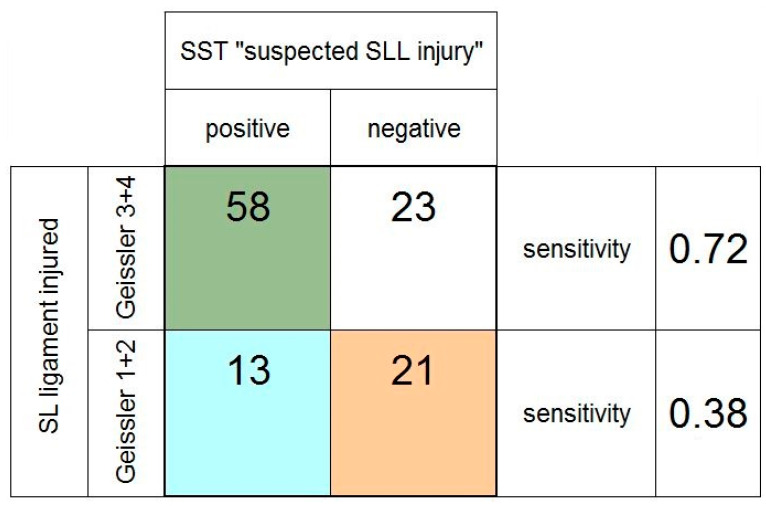
Fourfold table of the scaphoid shift test (SST) in the subgroup of patients referred specifically for suspected scapholunate (SL) ligament injury, depending on the severity of the SL ligament injury (grade 1 + 2 vs. grade 3 + 4, according to the Geissler classification).

**Figure 8 jcm-11-06322-f008:**
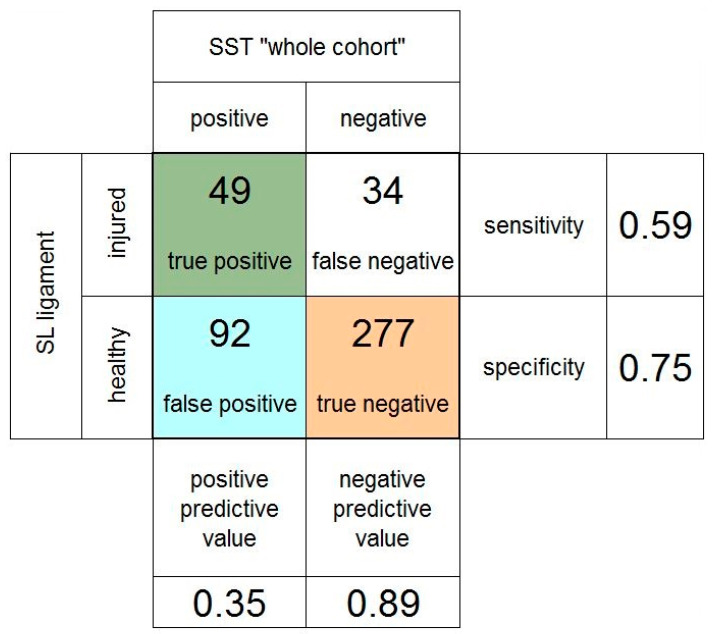
Fourfold table of the scaphoid shift test (SST) in the whole cohort of patients with static scapholunate (SL) ligament lesions defined as “SL ligament injured”, while no lesions or dynamic lesions during the arthroscopy were defined as “SL ligament healthy”.

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
