# Peer review of "Relevance of the Scaphoid Shift Test for the Investigation of Scapholunate Ligament Injuries"

_jcm, 2022, doi:10.3390/jcm11216322_

Round 1

Reviewer 1 Report

The present study is about the clinical relevance of an almost traditional clinical evaluation suspecting a ligamental lesion of the carpus. In a very large number of patients the authors found that Watsons's test is of relevance but might be misleading in lower graded pathologies. Although the study design is  retrospective, results are showing good evidence and flaws have been explained well. The conclusion is relevant for every clinical hand surgeon and should be incorporated in daily routine. Thank you for the effort and sharing the findings.

Author Response

Thank you very much for the feedback.

Reviewer 2 Report

Dear Authors, 

Thank you for this manuscript. I am a very big friend of clinical examination and one of the things I always try to teach my students is that the use of their hand to examine the patient is crucial for further treatment and development of a strategy in the meaning of "machine" examination like RX/MRI e.g.

Therefor I really appreciate your approach. 

There are some litte remarks that I would like to suggest to Improve your manuscript or at least for my better understanding. 

1) Is there really no need for an IRB Statement? Why you are useing patient data.

2) Can you provide a table of injury pattern? As this high number of cases would give a good insight to the reader. 

3) Could you please provide a picture series of the test step by step so that the audience will have a learning chance to know how it is done right. 

Best regards 

Author Response

There are some litte remarks that I would like to suggest to Improve your manuscript or at least for my better understanding. 

  • Is there really no need for an IRB Statement? Why you are useing patient data.

We conducted this study in accordance with the ethical guidelines of our institution. There was no need for an IRB statement, since we only conducted a chart review and evaluated data that have been gathered in the context of our quality management.

  • Can you provide a table of injury pattern? As this high number of cases would give a good insight to the reader. 

377 of 883 patients had a history of trauma to the wrist, the remaining 506 patients had no previous trauma to the wrist. All patients with previous trauma described a „fall to the extended hand“, a very typical trauma mechanism.

We therefore added the specification of this trauma mechanism in all patients with previous trauma in the manuscript (line 96)

  • Could you please provide a picture series of the test step by step so that the audience will have a learning chance to know how it is done right. 

Thank you very much for this excellent suggestion, we provided two pictures that illustrate the two steps of the scaphoid shift test (figures 2 A and B).

Reviewer 3 Report

Introduction:

1. Well written and enough background knowledge provided.

2. The purpose should be more specific to evaluate the specificity of scaphoid shift test (Watson test) as a confirmatory tool, since this study included the symptomatic patient undergoing arthrosopic surgery only (which implies those with highly suspicious of SLIL injuries)

Materials and Methods:

The method of this study is to check the clinical relevance of Watson test with the arthroscopic finding of SLIL, which considered the Gold standard in diagnosis of SLIL injury. The Achilles' heel of this study is that the number of true negative is fewer than the actual number leading to underestimate the specificity.

1. Line 83-84 please correct the number 1.134 and 1.103

2. Line 106 Please cite the original or modified Geissler classification to assess the SL injury arthroscopically  

3. Line 111 please provide the detailed IRB number of the institution

4. Line 123-133 Please explain the accuracy of checking SLIL via radiocapal joint only without assessing mid-carpal joint through MCR and MCU portal

Result: No specific comments to this section

Discussion:

1. Sensitivity means find the illness correctively from the illness. The explanation in Line 235 to 237 should be well interpreted.

2. Please put more explantation on the difference between definition of dynamic lesion as "SL ligament healthy" or not in the confusion matrix (Figure 3 vs Figure 7).

3. The included population is highly symptomatic patients with highly suspicious of SL injury. The true negative might be underestimated, which should be over-emphasis rather than stating negative test might need more work up in conclusion.

Author Response

  1. The purpose should be more specific to evaluate the specificity of scaphoid shift test (Watson test) as a confirmatory tool, since this study included the symptomatic patient undergoing arthrosopic surgery only (which implies those with highly suspicious of SLIL injuries)

Thank you for the suggestion, we added „…, and in particular the specificity“ to the text (lines 80 and 81).

Materials and Methods:

The method of this study is to check the clinical relevance of Watson test with the arthroscopic finding of SLIL, which considered the Gold standard in diagnosis of SLIL injury. The Achilles' heel of this study is that the number of true negative is fewer than the actual number leading to underestimate the specificity.

This is true. like mentioned in the discussion section (lines 253-255) most patients who had a negative result in the scaphoid shift test did not undergo arthroscopy, resulting in a possible bias of the negative predictive value and specificity.

  1. Line 83-84 please correct the number 1.134 and 1.103.

We corrected this.

  1. Line 106 Please cite the original or modified Geissler classification to assess the SL injury arthroscopically.

We cited the suggested reference.  

  1. Line 111 please provide the detailed IRB number of the institution

We conducted this study in accordance with the ethical guidelines of our institution. There was no need for an IRB statement, since we only conducted a chart review and evaluated data that have been gathered in the context of our quality management.

  1. Line 123-133 Please explain the accuracy of checking SLIL via radiocapal joint only without assessing mid-carpal joint through MCR and MCU portal

Thank you very much for the note. We have not described the arthroscopy detailed enough. We added the missing information with the sentence "Additional evaluation of the SLL was performed via MCR and MCU portals in 329 cases.” in lines 135 and 136.  

Result: No specific comments to this section

Discussion:

  1. Sensitivity means find the illness correctively from the illness. The explanation in Line 235 to 237 should be well interpreted.

We added in lines 246/247 the specification:  „…,since the sensitivity of the scaphoid shift test for the detection of low-grade lesions might be low.“

  1. Please put more explantation on the difference between definition of dynamic lesion as "SL ligament healthy" or not in the confusion matrix (Figure 3 vs Figure 7).

We added the following explanation to the text (lines 183/184):

„In figure 3 we defined dynamic, as well as static SL-lesions as „SL ligament injured“. However, in figure 7 we evaluated static lesions separately.“

  1. The included population is highly symptomatic patients with highly suspicious of SL injury. The true negative might be underestimated, which should be over-emphasis rather than stating negative test might need more work up in conclusion.

 This is true, the true negative patient might be underestimated, like mentioned under the limitations in the discussion section. In order to be more clear we added „…. in a symptomatic patient“ in line 271.

Round 2

Reviewer 3 Report

1. The IRB is always need, no compromise.

2. Please make sure the mid-carpal arthroscopy always done in your institute. There is a strong inter-observer bias in viewing subtle SL injury arthroscopically. As a result, a recent article published by the senior author David Ring [J Hand Surg Am. 2022;47(10):962e969] questioning the necessity of additional mid-carpal arthroscopy for diagnosis SLIL. Please clarify your standardized steps in diagnosing SLIL; otherwise, please make a statement in your limitation.

Author Response

Dear reviewer,

thank you for your comments on our manuscript.

1. Of course we involve our institutional review board in all studies including patient related data. However, in some instances no formal IRB number is assigned, because the study is sometimes interpreted as justified quality  control without a need for an extensive formal board review. We therefore cannot provide an IRB number, but can state the study has been performed after institutional board review.

2. You are right, we added the following sentence to the "limitations section": "Another limitation of our study is the fact that midcarpal arthroscopy was not routinely performed on all patients. We generally feel that visualization of the SLL is more reliable from the midcarpal joint space, but the retrospectice nature of this study makes it difficult to assess why in some cases only radoiocarpal arthroscopy was performed. It cannot be ruled out that some SLL injuries might have been missed, but considering the large total number of cases we do not think that this would alter the outcome of this study substantially."